# "I live with pain, it cannot go away": Lived experiences of childhood and adolescent pulmonary tuberculosis survivors — a qualitative study

Esin Nkereuwem[1,2⊙]*, Oluwatosin Nkereuwem[1⊙], Alpha Omar Jallow[1], James Owolabi[3], Assan Gibba[1], Fatoumatta S. Jawara[4], Zainab Manneh[4], Alex Opoku[4], Virginia Bond[5,6‡], Toyin Togun[1,2‡], Beate Kampmann[1,2,7‡]

**1** Vaccines and Immunity Theme, MRC Unit The Gambia at the London School of Hygiene and Tropical Medicine, Fajara, The Gambia, **2** Faculty of Infectious and Tropical Diseases, London School of Hygiene and Tropical Medicine, London, United Kingdom, **3** Department of Public Health, College of Health Sciences, Bowen University, Iwo, Nigeria, **4** School of Art and Sciences, University of The Gambia, Brikama, The Gambia, **5** Zambart, School of Public Health, University of Zambia, Lusaka, Zambia, **6** Department of Global Health and Development, Faculty of Public Health and Policy, London School of Hygiene and Tropical Medicine, London, United Kingdom, **7** Charité Centre for Global Health, Institute of International Health, Berlin, Germany

⊙ These authors contributed equally to this work.
‡ These authors also contributed equally to this work.
* esin.nkereuwem@lshtm.ac.uk

## Abstract

Childhood and adolescent tuberculosis (TB) survivors often face ongoing challenges even after completing their treatment. While the biomedical aspects of TB recovery are well studied, there is still limited understanding of how young survivors cope with the long-term impact of TB on their daily lives, development, and aspirations. Guided by an illness-narrative and phenomenological perspective, this qualitative study explored the lived experiences of children and adolescents who had completed treatment for pulmonary TB in The Gambia. Using a phenomenological approach, we purposively selected 33 participants from a larger longitudinal cohort study. Data were collected through participatory workshops that incorporated art-based methods and semi-structured in-depth interviews. Thematic analysis was used to identify patterns in participants' post-treatment experiences. Four main themes emerged: (1) persistent physical health challenges, such as fatigue and chest pain; (2) psychosocial difficulties, including stigma, fear of recurrence, and social withdrawal; (3) educational disruption and academic setbacks; and (4) evolving career aspirations, shaped by both limitations and newfound motivation. Participants' experiences varied by age and gender, with younger children relying on caregiver interpretations and older adolescents articulating complex emotional and identity-related reflections. Gender norms influenced the types of responsibilities and social roles participants attempted to resume. The findings highlight the multifaceted and long-term impact

**Data availability statement:** The qualitative data supporting this study include interview and group discussion transcripts that contain sensitive personal narratives from children and adolescents who survived tuberculosis. Making these data publicly available would violate the conditions approved by The Gambia Government/MRC Joint Ethics Committee and the LSHTM Observational/Interventions Research Ethics Committee, which prohibit open sharing of identifiable or potentially re-identifiable qualitative data. Anonymised transcripts may be shared upon reasonable request to the ethics committee or the Head of Data Management and Architecture, pending a formal review that ensures compliance with participant confidentiality and institutional data-sharing policies. Interested researchers can contact the ethics committee (ethics@mrc.gm) or the Head of Data Management and Architecture (information@mrc.gm) and are required to complete a data-sharing agreement. The data will remain stored and available for request at the MRC Unit The Gambia at LSHTM's Electronic Archive for at least 10 years following the conclusion of the research study.

**Funding:** This study was funded by the EDCTP2 Programme supported by the European Union (grant number TMA2020CDF-3197—Childhood TB Sequel to EN) and a Royal Society of Tropical Medicine and Hygiene (RSTMH) Early Career Grant to ON. The funders had no role in study design, data collection and analysis, decision to publish, or preparation of the manuscript.

**Competing interests:** The authors have declared that no competing interests exist.

of TB on young survivors. Recovery was experienced as a continuum from illness to post-treatment life, requiring rehabilitation-oriented support that integrates psychosocial counselling, school re-entry, and community-based stigma reduction. Integrating such support into post-TB care and public health programming is essential to improving outcomes for paediatric TB survivors.

## Introduction

Tuberculosis (TB) remains a significant global public health challenge and consistently ranks among the top ten causes of death [1]. The impact on children under 15 years is particularly concerning, as it is one of the leading causes of death from a single infectious disease [1]. Notably, at least 80% of childhood TB cases affect the lungs. Fortunately, treatment outcomes for children are generally favourable, with over 85% achieving treatment success after their first episode of TB [1,2].

Pulmonary TB has traditionally been considered successfully treated when a patient is declared cured or has completed their treatment [3]. However, this approach does not fully capture the clinical reality. Unlike many other respiratory infections, TB often leaves lasting damage to the lungs, which can turn the condition into a chronic, non-communicable health issue [4,5]. This highlights a significant oversight in our current classification of treatment outcomes, as they fail to acknowledge the enduring health effects that TB survivors experience. Increasingly, research suggests that these long-term consequences may be more common than previously recognised [6].

Recent studies have brought to attention the wide-ranging long-term effects of pulmonary TB in adults [7–13]. Even after completing treatment, many adult TB survivors continue to experience chronic physical health problems, which often lead to repeated hospitalisations [7,9,10,14]. Furthermore, there is growing awareness of the psychological and socioeconomic challenges that these individuals and their families experience [8,11–13].

These physical and psychosocial burdens associated with post-TB lung disease (PTLD) are also evident in children and adolescents. For instance, childhood pulmonary TB significantly increases the likelihood of having impaired lung function and is associated with a lower health-related quality of life [15]. Unlike adults, children and adolescents are in a critical stage of physical, cognitive, and social development. Disruptions caused by chronic illness during this period may have a compounding effect, influencing educational attainment, social integration, and long-term life opportunities [16]. These life-course implications highlight the importance of understanding how young TB survivors experience and adapt to life after treatment.

While several qualitative studies have explored the social and psychological dimensions of TB, most have focused on adults and on the treatment phase [11–13]. Far less is known about how children and adolescents navigate the period following treatment completion, when physical recovery coincides with evolving social identities and developmental transitions [17]. By centring on post-treatment experiences of

young TB survivors, this study expands our understanding beyond clinical cure and highlights the need for a comprehensive understanding of their social experiences, which can inform approaches to addressing PTLD in these populations.

This study aims to explore the perceptions and lived experiences of these survivors, seeking to gain an in-depth, 'emic' understanding of the social dimensions of life after the completion of TB treatment. We employed qualitative methods informed by phenomenological principles, which are particularly well-suited to exploring questions related to meaning, identity, and experience from the perspectives of pulmonary TB survivors [18,19]. Our goal was to understand how survivors of childhood and adolescent pulmonary TB perceive and articulate the various dimensions of their post-TB experiences and how these experiences impact their daily lives. This research was conceptually guided by the illness-narrative perspective, which views illness as both a biological and social phenomenon that shapes identity, meaning, and relationships [20,21]. Framed within a phenomenological approach, we examined how children and adolescents reconstruct their sense of self and normalcy after TB treatment [22].

## Materials and methods

### Study setting and design

The Gambia, a West African country with a high TB burden, reports nearly 3,000 TB cases annually, with around 8% occurring in children under 15 years [1]. This qualitative study was embedded within the *Childhood TB Sequel*, a prospective cohort study conducted in the Western I and II Health Regions of The Gambia – QUAN(qual) design. The primary aim of the *Childhood TB Sequel* is to characterise the multidimensional sequelae of pulmonary TB among children and adolescents [23].

Between April 1, 2022 and July 1, 2024, the main cohort study enrolled 100 participants, followed them for 12 months after their treatment, collecting data at three different time points across clinical and functional domains [23]. To explore the lived experiences of our study population, we used an interpretative phenomenological approach (IPA), which is particularly suited to understanding how individuals make sense of significant life experiences, such as illness and recovery [22]. This participant-led approach enabled a detailed and nuanced understanding of the post-TB phenomenon.

### Study team

The research team comprised a multidisciplinary group of social scientists and a clinician with expertise in qualitative research. The lead researchers, EN and ON, jointly designed and facilitated data collection and analysis, contributing their unique perspectives from their respective backgrounds. Additionally, EN was involved in the broader *Childhood TB Sequel* project and was familiar with the study participants, which helped to provide continuity between the qualitative and quantitative components of the research. All team members had lived in The Gambia for over six years and had a deep understanding of the health system and the sociocultural context, thus eliminating the need for a separate orientation to the setting. Reflexivity practices were utilised throughout the project and are described below.

A child protection specialist, who was independent of the research team, was present during the group discussions and on standby during the in-depth interviews (IDIs). Their role was to safeguard the welfare and rights of participants and ensure they were protected from any potential abuse and exploitation.

### Participant recruitment and sampling

Participants who attended any of the three *Childhood TB Sequel* study visits were eligible to participate in the qualitative arm of the study. The *Childhood TB Sequel* included children and adolescents aged 19 years and below who had completed treatment for drug-sensitive pulmonary TB (DS-TB) with an outcome of *cured* or *treatment completed*. Specifically, we invited children and adolescents aged ten years and above, as well as caregivers of children below ten years, to participate in the qualitative study.

Participants were purposively sampled to ensure a diverse representation of the *Childhood TB Sequel* study population, including a range of demographics, disease severity, and treatment experiences. "Treatment experience" in this context included illness severity and whether participants required hospitalisation or missed school due to TB. This approach allowed for the intentional selection of individuals who could provide in-depth insights into the varied outcomes and long-term effects of TB. This ensured that the findings would capture the full spectrum of experiences relevant to the study's objectives. The participants were approached in person during their study visits or contacted by phone to participate in the qualitative study.

The study sample included participants aged 6–19 years, with an almost equal number of males and females. Most participants had completed their TB treatment within the last 6–12 months at recruitment. Caregivers in the study were primarily mothers or other close family members of younger children, allowing for perspectives from both children and adolescents at different developmental stages. Although some eligible participants could not be contacted or were unavailable due to scheduling conflicts, none declined due to dissatisfaction with their TB care experience.

## Data collection

We collected data using facilitator-guided group discussions during participatory workshops and in-depth interviews (IDIs).

## Group discussions

We held three participatory workshops between April 27, 2024 and May 11, 2024, during which facilitator-guided group discussions were conducted. The workshops were held separately for caregivers of children aged under 10 years, young adolescents aged 10 to under 15 years, and adolescents aged 15 and above (see S1 Table). These workshops aimed to facilitate discussions among the childhood TB survivors and their caregivers, allowing them to share common experiences and express their observations of the TB and post-TB experience.

We invited between 10 and 20 caregivers, children, and adolescents from each group to participate in the workshops. Caregivers were encouraged to accompany the children and younger adolescents. The workshops were conducted in a neutral location with a dedicated play area for the children to ensure a comfortable and relaxed environment. This neutral location was a rented community hall that provided both indoor and shaded outdoor spaces. There were breaks between the activities, and participants and caregivers were provided with refreshments and transportation refunds. The child protection specialist was available to provide mental health first aid and coordinate referrals for psychological interventions where needed.

The study team present during each workshop comprised a female social scientist (ON) with an MPH experienced in conducting group discussions among children and adolescents, a male MD (EN) with training and experience in designing and conducting qualitative research, and six research assistants (two female and four male) who all hold bachelor's degrees in social science or public health and have field experience in conducting group discussions and interviews.

During each workshop, the discussions were guided by a lead facilitator (ON) and the social science research assistants. The group sessions began as a large group with some icebreakers and introductions before the participants were invited to engage in age-appropriate activities in small groups, followed by individual and group debrief sessions. Each smaller group activity was facilitated by a pair of social science research assistants and had a similar structure (see S1 Text). These activities included art-based research methods (collages, body mapping, and drawing), games, role-playing, and storytelling [24]. The activities during these workshops were guided by a flexible discussion guide designed to address the study's objectives.

Participants were given materials such as magazines, newspapers, scissors, and colourful pens to create individual collages and drawings that represented their past experiences, present selves, and future aspirations. The facilitators introduced the method to encourage storytelling and interaction among participants. After creating their collages, participants shared and discussed them with the group, using the collages as a means to reflect on their lives and experiences.

During the body mapping exercise, participants gathered around a large outline of the human body and were encouraged to colour the areas where they felt symptoms, pain, or discomfort. The end result was a collective image that vividly illustrated the locations of discomfort experienced by participants through different colours.

The workshops were audio-recorded using encrypted devices, and the researchers took notes to capture the highlights of the conversations, as well as any nonverbal cues, during the discussions. The workshop lasted approximately two hours for caregivers of children aged 5 to under 10 years, about three hours for young adolescents aged 10 to under 15 years, and three hours for adolescents aged 15 years and above. The workshops were moderated in English and accompanied by one of the research assistants proficient in the commonly spoken local languages, who served as an interpreter. Most children and adolescents spoke basic English, but interpretation into *Mandinka*, *Wolof*, or *Fula* was provided when needed. Data were transcribed in English following a single-step translation process directly from the local language during transcription.

### In-depth interviews (IDI)

Following each age group's workshop, IDIs were conducted with purposively selected children, adolescents, and caregivers who had participated in the workshops. Four primary caregivers of children aged under ten, four young adolescents aged ten to under 15 years, and four adolescents aged 15 years and older were invited for the IDIs. Participants were selected based on the richness of their engagement in the workshops, the diversity of their experiences, and their willingness to share personal stories in greater detail. The interviews were conducted between May 6, 2024 and June 26, 2024.

The social science field workers (AOJ, JO, AG, FSJ, ZM, and AO) conducted the IDIs in either English or the local language preferred by the participants. The interviews were conducted face-to-face with four caregivers and 11 adolescents. A semi-structured interview guide, developed from a previous qualitative study conducted among adolescent and adult TB survivors in The Gambia, was used (see S1 Text) [25]. The guide was flexible, allowing for revisions based on insights gathered from group discussions and earlier IDIs, including emerging themes and ideas. The questions in the guide were intentionally broad, with additional prompts to encourage respondents to share their experiences in depth.

Each interview was audio-recorded using an encrypted device, with each session lasting approximately 35 minutes on average. We continued the interviews until data saturation was reached, meaning that no new themes emerged. On average, the IDIs were conducted between one and seven weeks after the group workshops. There were no repeat interviews.

### Data analysis

The group discussions and interviews were recorded using an encrypted device, and the researcher took notes to capture any nonverbal cues. Participants were individually debriefed about their collages, and these sessions were also recorded. Research assistants made detailed notes during the participatory workshops to highlight key conversations. Research assistants performed verbatim transcriptions of the audio recordings independently in pairs. A third person assessed the transcripts for similarity and accuracy. The information obtained from the group discussions was used to refine the structured guides for the IDIs.

We used an inductive approach to generate themes and subthemes [26]. Initially, two researchers familiarised themselves with the transcripts to analyse the data. They independently coded a subset of the data before collaboratively refining their coding approach and impressions of the emerging themes. The initial coding was performed manually using Microsoft Word and Excel by AOJ and JO.

To enhance trustworthiness, the entire research team reviewed the organisation of the codes and provided additional input. Both researchers then coded the remaining transcripts independently, organising the codes into subthemes using NVivo 12 software (QSR International, Melbourne, Australia), while considering the context to inform their thematic

interpretation. Next, all researchers collaboratively refined the interpretive decisions. Interpretive discussions were held during team meetings to agree on theme structures, resolve discrepancies, and ensure contextual coherence.

### Researchers' perspectives and reflexivity

Before conducting the workshops, the study team met regularly to discuss and document their personal and professional perspectives, as well as their initial impressions of the data collection and interview process. This practice continued after each workshop and throughout the interviews and data analysis. The coders also participated in these reflexive practices to ensure an open, honest, and reflective approach to generating themes and sub-themes. The two lead researchers (EN and ON) shared a common goal of gaining a deeper understanding of the qualitative dimensions of post-TB lung health and identifying the participants' experiences. This research interest was communicated to the participants.

EN is a clinician–researcher with long-standing engagement in childhood TB research in The Gambia, while ON is a social scientist with expertise in qualitative research with children and adolescents. Their complementary perspectives enabled an approach that balanced clinical insights with sensitivity to the social context of participants' lives. Their previous work in TB research also helped them build rapport with participants. Reflexive discussions held during and after data collection allowed them to consider how their backgrounds and assumptions might influence interpretation. This reflexive awareness strengthened the credibility and trustworthiness of the study's findings.

### Ethical considerations

For the qualitative study, we obtained separate written informed consent from adolescents 18 years and older, as well as from the caregivers of participants under 18 years. Adolescents aged 12–17 years provided written assent, while those under 12 years gave verbal assent, in accordance with local ethics guidelines. Additionally, if any data was collected from the caregivers of participants aged 10–19 years, consent was also obtained from those caregivers. Participation in the qualitative study had no impact on the individual's involvement in the *Childhood TB Sequel* study.

Ethical approval for the qualitative study was obtained from The Gambia Government/MRC Joint Ethics Committee (Reference 28229) and the LSHTM Observation/Interventions Research Ethics Committee (Reference 28229).

This study followed the Consolidated Criteria for Reporting Qualitative Research (COREQ) to ensure transparency and rigour in qualitative reporting (see S2 Table) [27].

### Inclusivity in global research

Additional information regarding the ethical, cultural, and scientific considerations specific to inclusivity in global research is included in the Supporting Information (S2 Text)

## Results

### Characteristics of the sample

The recruitment of participants is summarised in Fig 1. Twenty-five caregivers, ten adolescents aged 10 to <15 years, and 16 adolescents aged ≥15 years and older participated in the group discussions. The caregivers were predominantly mothers (n = 11), with a median age of 40.0 years (IQR 23.0 to 46.0). The younger group of adolescents consisted of six boys and four girls, with a median age of 12.5 years (IQR 11.8 to 13.1), while the group of adolescents aged 15 years and older included seven boys and nine girls, with a median age of 17.8 years (IQR 16.7 to 18.1).

At the time of qualitative data collection, most participants had completed TB treatment within the past year. This timing allowed us to explore both recent and ongoing post-treatment experiences while memories of the illness and care remained vivid.

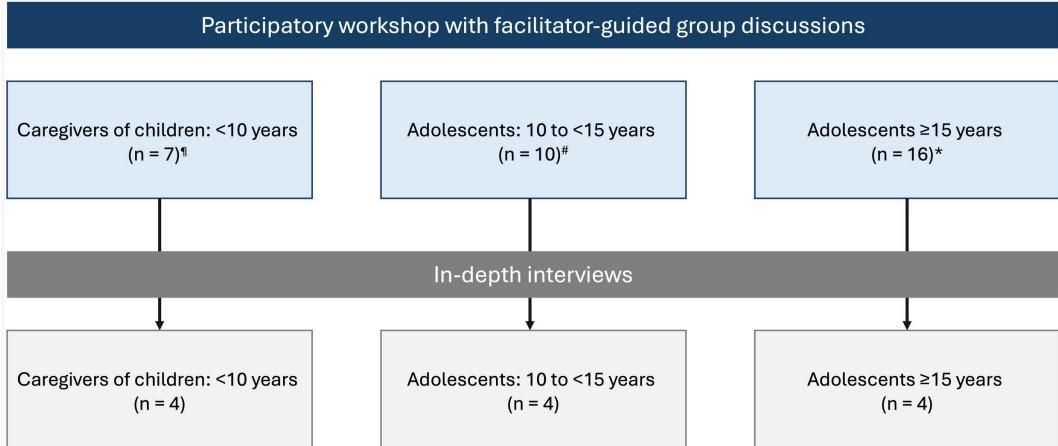

**Fig 1. Participants in the research.** *Six mothers and one father; #Nine caregivers were also present at this workshop (two mothers, four fathers, one grandmother, and two siblings); †Nine caregivers were also present at this workshop (three mothers, two fathers, and four siblings).*

The qualitative findings revealed a complex interplay of physical, psychological, educational, and aspirational changes in participants' lives following TB treatment. Although the study focused on life after treatment, participants often referenced their experiences during the initial stages of illness and treatment, which strongly influenced their understanding of recovery and present challenges. Consequently, the themes below represent a continuum of experiences across diagnosis, treatment, and post-treatment, rather than separate, isolated stages.

To aid interpretation, all quotations in the Results section are identified by participant type and age (e.g., "17-year-old male," "caregiver of an 11-year-old female") to highlight age-related and role-specific perspectives. The themes that emerged reflect how survivors and their caregivers processed the ongoing impact of TB, made sense of their experiences, and navigated altered life trajectories (Table 1).

## Physical health challenges

Participants and caregivers consistently described physical limitations as a defining feature not only of life after treatment but also as lingering reminders of the illness and its treatment. Their narratives linked bodily weakness during TB to ongoing constraints that shaped identity, social engagement, and household functioning.

**Physical activity limitations.** Participants commonly experienced a significant reduction in their ability to engage in physical activities, particularly in sports and exercise. Many, especially adolescent boys, described sports and play as important aspects of their identity and daily routine. They often felt weaker and less capable, which hindered their participation in activities they once enjoyed, leading to frustration and a sense of loss.

> *"TB affected my sports life. This year, I could not participate in inter-house sports. When I ran a short distance, I easily got tired. The strength that I had for sports before is completely different now"* (17-year-old male).

Younger participants echoed this disruption of play as a core childhood activity. Their frustration, along with their adaptation to new physical limitations, reflects an early and ongoing negotiation with bodily constraints.

> *"Before I got sick, I usually played football normally. But when I got sick, I wanted to play football, but I could not. When I'm not feeling well, when my people are running at the field, I tell my coach that I cannot run… If my chest is hurting,*

**Table 1. Themes and sub-themes that emerged from the discussions and interviews.**

| Themes | Sub-themes |
|---|---|
| Physical health challenges | • Physical activity limitations<br>• Persistent symptoms<br>• Challenges in performing daily chores<br>• Impact on recreational activities |
| Psychosocial challenges | • Emotional and social support<br>• Triggers and memories<br>• Self-esteem and isolation<br>• Secrecy and disclosure |
| Disrupted education and academic setbacks | |
| Shifting career aspirations | |

*then I will sit down. When it's time to play, he will put me inside to play. Even at that, after a while, I sit"* (9-year-old male).

Through the visual body-mapping activity, children were able to collectively illustrate the pervasiveness of their physical symptoms (Fig 2). Many marked the chest area in red or orange, conveying sensations of discomfort, while blue and purple tones denoted fatigue or weakness. Frequent markings around the ribs and upper back indicated continued breathlessness and muscle pain. Together, these images reflected the children's shared awareness of their bodies as sites of persistent strain and incomplete recovery.

**Persistent symptoms.** Chronic pain and fatigue were described as constant features of daily life. Participants did not view these symptoms as isolated incidents but as part of a new normal, a residual identity shaped by the illness. Their narratives reflected both acceptance and ongoing distress.

*"Now it's better, but the pain is still there. Sometimes I tell my sister I live with pain. I think this pain is part of me now. It cannot go away. Yeah, but still, you just have to push yourself"* (18-year-old female).

Caregivers corroborated this ongoing struggle, expressing concern over relapses or unresolved symptoms often manifesting as anxiety about the child's future health.

*"The recurrent cough and breathlessness worry me a lot"* (caregiver of a 12-year-old male).

**Challenges in performing daily chores.** Adolescents, particularly girls, spoke of a diminished ability to carry out household tasks, which they associated with reduced productivity and changes in family roles. This situation added an extra layer of frustration and guilt to their recovery narratives.

*"What has changed in her life is that her productivity has reduced, in terms of the work that she does. She can't do her work as she wishes"* (caregiver of an 11-year-old female).

*"For me … I do cooking, I do laundry sometimes, but most of the time if I do it, then I will have chest pain"* (18-year-old female).

The struggle with everyday chores signified a disruption of routine and autonomy, reflecting a broader sense of being held back by their post-TB bodies.

**Impact on recreational activities.** Across different age groups, reduced engagement in leisure activities became a source of social withdrawal. Play and recreation, typically experienced in the company of peers, were no longer effortless or enjoyable, leading to emotional isolation.

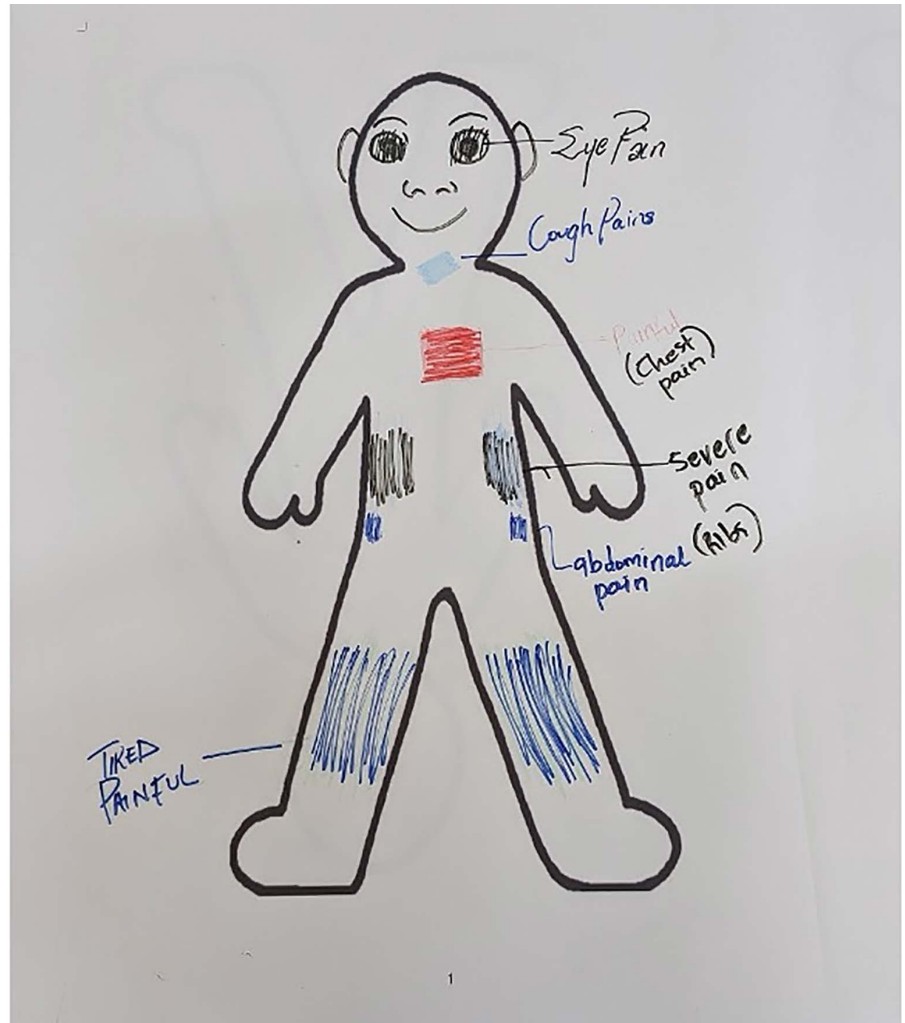

**Fig 2.** **Body mapping created by participants showing the areas of predominant physical symptoms they experienced (facilitatory workshop, May 2024).**

*"Before I got sick, I usually played football normally, but when I got sick, I wanted to play football, but I could not"* (9-year-old male).

*"When I am playing, I get easily tired. Sometimes, when my friends call me to play with them, I would sometimes respond no"* (11-year-old female).

*"Sometimes he will be at one place, even if other kids are playing, he will not participate. He can't go for football, sometimes he will be sitting alone and feeling sad"* (caregiver of an 11-year-old male).

Notably, some participants felt fully recovered and unaffected by their previous illness, highlighting differences in recovery and the possibility of complete symptom resolution. Taken together, these accounts suggest that lingering symptoms can slow re-engagement in important activities, with downstream effects on confidence and social connection.

## Psychosocial challenges

The psychosocial burden of TB extended beyond the treatment period, with many adolescents describing their post-TB emotions as continuations of the fear, stigma, and isolation experienced during the illness. These reflections reveal how the psychological aftermath of TB blurs the boundary between the treated and the still-recovering self. This effect was felt most strongly by older adolescents, many of whom described significant internal changes. For them, the illness became not just a physical challenge but also a transformative emotional and relational experience.

**Emotional and social support.** Support from mothers, family members, and occasionally teachers emerged as a protective factor during illness. The participants expressed deep gratitude, indicating a strong awareness of their interdependence during these challenging moments. In this context, recovery involved not only physical healing but also the strengthening of relationships.

*"During the period of my sickness, my parents, both of them, and the doctors helped me. I'm grateful to them for their help"* (12-year-old male).

*"When I was sick, I did not think I would get better. Now, I am on my feet, going to work and doing, thanking my mum, who also supported me in everything. I was even saying that all my neighbours who thought I would not get better, I'm now better"* (18-year-old female).

*"The time I was sick, my classmates, all of them, came and visited me. Every time they would come. Some of my teachers would call and ask me: 'So, how are you feeling now?' I would say I'm fine. 'We miss you in school.' I wished to attend school, but I was not strong enough"* (18-year-old female).

These statements convey more than simple gratitude. They emphasise how care and affirmation played a crucial role in restoring a sense of social belonging during vulnerable times.

**Triggers and memories.** Recovery was often interrupted by involuntary reminders, such as phone calls, school visits, or health checks, which reactivated fear and sadness. These moments indicate unresolved trauma and a sense of uncertainty about the future.

*"I used to think the disease would return. I feel like it will come back, and it gets me scared"* (17-year-old male).

*"When I was done with the treatment, when everything was clear, they told me I was negative. I burnt [my treatment card] immediately... So, after some weeks, I got a call to come back (for the research). That day, I don't know how I was feeling. Many things were going through my mind. I felt sad"* (18-year-old male).

**Self-esteem and isolation.** Psychological distress, particularly among male adolescents, emerged as a significant issue. Some participants voiced experiences of depression, existential thoughts, and a deep sense of loss regarding their social identity.

*"It has affected the way I think. I think about death a lot... I am usually alone. No one visits me. People run away from me, so I see myself as somebody who is dead. That disturbs me a lot psychologically"* (17-year-old male).

*"I like to sit and chat with my friends, but I can't"* (13-year-old male).

These comments highlight the emotional toll of stigma and isolation, with some participants internalising feelings of exclusion, equating it to a form of symbolic death.

**Secrecy and disclosure.** Tuberculosis (TB) continued to be associated with stigma, even after individuals were cured. Many adolescents often chose to remain silent about their illness instead of disclosing it, fearing rejection and

misunderstanding from others. Their decisions reflected a careful trade-off between being true to themselves and protecting their relationships.

> "Only my mum and my sister knew. I didn't feel like telling others because maybe they would distance themselves from me" (18-year-old female).

> "None of my friends knew I was sick. Among my family, only my parents knew, not even my younger siblings" (17-year-old female).

These personal accounts reflect broader societal norms regarding illness, visibility, and self-worth. The need to hide a TB history highlights its enduring significance in everyday relationships.

### Disrupted education and academic setbacks

Education was not simply interrupted; it became a space where illness, identity, and opportunity collided. Participants frequently referred to missed school days during treatment as the foundation for later academic decline, underscoring how the educational consequences of TB unfold across the treatment and post-treatment continuum. Participants' academic performance was deeply tied to their sense of self-worth and future outlook.

> "When I go to school, I don't concentrate" (16-year-old female), while another reflected:

> "My school performance has also dropped. From grade one to five, I was in the first position. Five to six grades, I dropped to the second position" (17-year-old male).

> "She is someone who tries a lot. But now her performance at school is reducing... because of her health" (caregiver of an 11-year-old female)

Participants linked cognitive decline and academic struggles to a mix of fatigue, missed school days, and emotional distress. Many described having to repeat grades due to prolonged absences.

> "I got TB when I was in grade eight. I wasn't going to school at the time TB was disturbing me. After my treatment, I decided to go to school but I was told that I should repeat grade eight" (12-year-old male).

School responses (e.g., grade repetition, informal exemptions) often addressed attendance rather than learning loss, reinforcing gaps that later appeared as reduced motivation.

> "Sometimes he does not go to school because walking long distances disturbs him. Sometimes when he had an 'attack' (exacerbation of symptoms) at school, the teacher will allow him to go home to take his medication" (caregiver of a 12-year-old male).

These academic setbacks often translated into emotional challenges. Participants expressed feelings of being left behind and disconnected, especially when their classmates advanced.

> "I feel left behind. I was so behind... They were all going to school, I'm staying at home" (18-year-old female).

> "It affected my schooling because my schoolmates, whom I was on the same level with, all left me behind" (12-year-old male).

These accounts underscore how the disruption of education contributed not only to cognitive loss but also to social disconnection and decline in self-confidence. Thus, academic setbacks were not only individual consequences of illness but products of the interaction between recovery pace and school routines.

### Shifting career aspirations

The experience of TB had a profound impact on career aspirations, influencing the participants' future plans and ambitions. For some, the illness dampened their enthusiasm for their original career goals, while others found renewed purpose in pursuing a career in the healthcare field.

The physical and emotional challenges participants faced during their illness led many to reassess and shift their career aspirations. Some lowered their ambitions or opted not to pursue further education or professional goals, reflecting the significant impact on their future plans:

*"I wanted to be an accountant... and I wanted to do football, but since my TB came, all that was away. I cannot do sports now because I usually get tired easily, and breathing becomes a problem when I jump or sometimes run. My breathing is a problem, especially my ribs—they disturb me a lot. My mum says it's not healthy, so it's better I focus on being an accountant"* (18-year-old female).

*"I really wanted to be a nurse, but after I got sick, I lost all hope. I even dropped out of school"* (18-year-old female).

Some participants experienced uncertainty and setbacks during their illness, leading them to lose enthusiasm for their initial career plans. The physical limitations and emotional toll of TB caused them to reconsider or abandon their original ambitions.

*"Although I don't use to pass science that much, I always used to tell my mum that one day, I will be a doctor... but the time I was sick, I gave up on that"* (18-year-old female).

Conversely, for a few participants, their experience of dealing with TB strengthened their resolve to enter the healthcare field. The desire to help others who might be suffering from similar health issues became a motivating factor in their career choices. These reflections stand out as deviant cases, where illness served as a source of motivation rather than loss, prompting a renewed commitment to pursue medical or nursing careers.

*"Even though it's still affecting me somehow, I still want to be a nurse. I don't think this will stop me from doing that. I want to help those who get TB also"* (18-year-old female).

Another participant, who had once hoped to return to sports, ultimately decided to focus on their education and future career, stating, *"I lose hope in that area... I am focusing more on my education now"* (17-year-old female).

The collage activity revealed these changes in aspirations, as participants used visual metaphors to express their grief, resilience, and hope for the future.

## Discussion

This study provides a nuanced understanding of the lived experiences of childhood and adolescent TB on survivors in The Gambia, revealing how the disease impacts not only physical health but also psychosocial well-being, educational attainment, and career aspirations. The findings are significant in that they document the long-term consequences of TB during formative developmental stages and illustrate how children and adolescents make sense of their illness through narratives of pain, resilience, and hope.

Four overarching themes emerged: persistent physical symptoms, psychosocial challenges including stigma and isolation, educational disruptions, and shifting aspirations. Participants described ongoing fatigue, chest pain, and breathlessness, which affected their ability to engage in physical activities and routine chores. These experiences were deeply intertwined with emotional distress and altered social roles. Stigma led to secrecy and withdrawal, while frequent absences from school undermined academic confidence and led to grade repetition. Despite these challenges, some participants found strength in their experience and expressed renewed aspirations, particularly toward healthcare careers.

Our findings are consistent with existing literature from LMICs, which shows that TB survivors, particularly adults, experience prolonged physical and mental health issues after treatment [6,9,10]. However, our study provides important contributions specifically focused on children and adolescents, highlighting how these challenges manifest during critical developmental stages and intersect with social identity, educational performance, and future aspirations [15,16]. The persistence of physical limitations among participants, especially symptoms like fatigue and breathlessness, has also been reported in prior studies [15,28]. This study expands on those findings by illustrating how these symptoms impact the roles and responsibilities of children within their families and communities.

Psychosocial outcomes in our study, such as feelings of shame, self-isolation, and emotional withdrawal, resonate with previous research on HIV and sickle cell disease in young populations [29,30]. Yet, in contrast, our participants exhibited a notable reluctance to disclose their illness, even to close peers and siblings. This indicates a specific stigma associated with TB that creates a significant barrier to social reintegration and highlights the need for targeted intervention [31]. This difference suggests that, while chronic illnesses broadly impact youth psychosocially, the sociocultural narratives surrounding TB may intensify feelings of deviance and secrecy. Nevertheless, the ability of young survivors to transform their adversity into motivation demonstrates their capacity to find meaning and purpose in their experiences, despite facing considerable social stigma [32,33].

TB significantly affected the educational and career paths of participants in various ways. Some participants experienced academic decline and loss of motivation, while others felt inspired to pursue careers in health-related fields. The role of family, teachers, and peers was pivotal in shaping these responses. Where social support was present, participants reported greater resilience and optimism [34]. These contrasting outcomes underscore the importance of providing targeted support for school re-entry and psychosocial services to help young survivors navigate the transition from illness to recovery and set future goals [35–38].

Our use of participatory visual methods complements similar work by Mackworth-Young et al. [24,39], where techniques such as collages and body mapping illuminate inner experiences that are often overlooked in traditional interviews. These tools facilitated the expression of feelings of grief, isolation, and hope in a way that is culturally meaningful and suitable for their developmental stage.

Participants' experiences were shaped by their age and gender roles, highlighting important differences within the group. Younger children tended to rely more on narratives from their caregivers, while adolescents provided deeper insights into their identities and mental health. Additionally, gender roles significantly influenced how individuals adjusted after illnesses like TB [40]. Girls frequently expressed pressure to resume domestic responsibilities, whereas boys grappled with the loss of athletic participation and social interactions. These differences highlight the necessity for recovery support systems that are both age-appropriate and gender-responsive, acknowledging the distinct expectations and challenges faced by children and adolescents during their recovery journeys.

To address the challenges identified, we propose integrating psychosocial counselling and structured educational reintegration into post-TB care protocols. Establishing school-based liaison programmes and peer support networks for young survivors could help mitigate the academic and social consequences. Additionally, public health messaging should be tailored to dispel myths and reduce stigma at the community level, particularly among peers, teachers, and extended families.

In the Gambian context, these insights point to several feasible, low-cost interventions that could strengthen post-TB care. Integrating school-based reintegration and peer-support initiatives can help young survivors regain confidence and continuity in their education, while community-led awareness efforts could reduce stigma and promote supportive environments. Embedding simple psychosocial counselling and routine symptom screening into post-TB follow-up would further align with a rehabilitation-oriented framework, ensuring both medical and social recovery. Together, these coordinated measures offer a practical, resource-sensitive approach to improving long-term outcomes for paediatric TB survivors.

### Strengths and limitations

The strengths of the study include its use of various qualitative methods, such as group discussions, IDIs, and visual data. The research team also employed a diverse sampling approach that included participants of different ages, and they demonstrated a strong contextual understanding of the subject matter. The triangulation of data sources enhanced the robustness of the findings.

However, there were some limitations, including potential bias from caregivers when representing the experiences of younger children. Additionally, there were challenges in translating local language expressions into English. The wide age range of participants (6–19 years) posed challenges for maintaining homogeneity, as recommended in idiographic analysis [41]. Nonetheless, this decision was intentional, as it allowed the exploration of developmental differences and age-specific experiences. While the findings may not be generalisable, the depth of contextual insights offers valuable transferability to similar high TB burden settings [42].

Reflexivity practices integrated into the analysis strengthened the interpretive process and acknowledged the researchers' professional and personal positioning. We consider the interpretive nature of IPA not as a bias but as a critical asset that facilitated a deeper understanding of meaning-making in young TB survivors [22].

### Conclusions

This study provides valuable insights into the lasting effects of TB on the lives of children and adolescents. These findings emphasise the need for comprehensive post-treatment care that includes support for physical health, emotional well-being, and education. Future research should examine longitudinal outcomes of recovery and evaluate interventions aimed at reducing stigma and enhancing resilience. Additionally, cross-context comparative studies could further elucidate how health systems and cultural norms shape the post-TB experiences for young people. To support young people affected by TB, clinicians should provide routine post-TB follow-up for unresolved symptoms and psychosocial concerns, educators should enable smooth school re-entry and academic support, and policymakers should embed stigma reduction and rehabilitation in national adolescent health policies.

### Supporting information

**S1 Table. Workshop programme.**
(PDF)

**S2 Table. COREQ checklist.**
(PDF)

**S1 Text. Interview guide.**
(PDF)

**S2 Text. Inclusivity in global research.**
(DOCX)

### Acknowledgments

We thank the children and their parents for participating in this study. We acknowledge the *Childhood TB Sequel* study team.

### Author contributions

**Conceptualization:** Esin Nkereuwem, Oluwatosin Nkereuwem, Toyin Togun.

**Data curation:** Esin Nkereuwem, Oluwatosin Nkereuwem, Alpha Omar Jallow, James Owolabi, Assan Gibba, Fatoumatta S. Jawara, Zainab Manneh, Alex Opoku.

**Formal analysis:** Esin Nkereuwem, Oluwatosin Nkereuwem, Alpha Omar Jallow, James Owolabi, Assan Gibba.

**Funding acquisition:** Esin Nkereuwem, Oluwatosin Nkereuwem.

**Investigation:** Oluwatosin Nkereuwem.

**Methodology:** Esin Nkereuwem, Oluwatosin Nkereuwem, Virginia Bond, Toyin Togun, Beate Kampmann.

**Project administration:** Esin Nkereuwem, Oluwatosin Nkereuwem.

**Resources:** Oluwatosin Nkereuwem.

**Supervision:** Esin Nkereuwem, Virginia Bond, Beate Kampmann.

**Visualization:** Esin Nkereuwem.

**Writing – original draft:** Esin Nkereuwem, Oluwatosin Nkereuwem.

**Writing – review & editing:** Esin Nkereuwem, Oluwatosin Nkereuwem, Alpha Omar Jallow, James Owolabi, Assan Gibba, Fatoumatta S. Jawara, Zainab Manneh, Alex Opoku, Virginia Bond, Toyin Togun, Beate Kampmann.

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
