## [Decision Letter · Decision Letter 0]

10 Sep 2025

PGPH-D-25-01458

"I live with pain, it cannot go away": a qualitative study exploring the lived experiences of childhood and adolescent pulmonary tuberculosis survivors

Dear Dr. Nkereuwen,

Thank you for submitting your manuscript to PLOS Global Public Health. After careful consideration, we feel that it has merit but does not fully meet PLOS Global Public Health’s publication criteria as it currently stands. Therefore, we invite you to submit a revised version of the manuscript that addresses the points raised during the review process.

I invite you to consider the comments and suggestions that the reviewers have provided. Regarding representing the quotes in a table as suggested by one of the reviewers, the authors should consider if that is the best way for representing their data.

We look forward to receiving your revised manuscript.

Kind regards,

Ferdinand C Mukumbang, PhD

Academic Editor

Journal Requirements:

2. In the online submission form, you indicated that “Anonymised interview transcripts are available from the corresponding author on request.”.

3. Uploaded as supplementary information.

Additional Editor Comments (if provided):

Reviewer #1:

Reviewer #2:

Reviewers' comments:

Reviewer's Responses to Questions

**Comments to the Author**

1. Does this manuscript meet PLOS Global Public Health’s publication criteria?

Reviewer #1: Yes

Reviewer #2: Yes

2. Has the statistical analysis been performed appropriately and rigorously?

Reviewer #1: N/A

Reviewer #2: I don't know

3. Have the authors made all data underlying the findings in their manuscript fully available (please refer to the Data Availability Statement at the start of the manuscript PDF file)?

Reviewer #1: Yes

Reviewer #2: No

4. Is the manuscript presented in an intelligible fashion and written in standard English?

Reviewer #1: Yes

Reviewer #2: Yes

Reviewer #1: 1. consider to rephrase the title for more clarity

“I live with pain, it cannot go away”: Lived experiences of childhood and adolescent pulmonary tuberculosis survivors — a qualitative study"

2. Introduction:

- Consider constructing the theoretical framework or perspective guiding the qualitative inquiry (e.g., illness narrative, trauma theory, etc.).

- Clarify how this study builds upon or diverges from existing qualitative TB studies.

3. Methods:

- Provide more information on the demographic characteristics of participants (age ranges, gender distribution, time since treatment completion, etc).

- Describe researcher positionality to enhance trustworthiness.

- Clarify if any software (e.g., NVivo, Atlas.ti) was used for coding.

4.Results:

- Ensure each theme is analytically developed, not just descriptive.

- Discuss any deviant or contradictory cases if applicable.

5. Discussion:

- Elaborate on how findings may inform rehabilitation frameworks or public health interventions.

- Include comparative findings from other global regions.

Conclusion:

- Summarize actionable recommendations for clinicians, educators, and policymakers.

References:

Check all references to comply with journals referencing guidelines

Reviewer #2: At the outset, I must congratulate the authors for the well structured and conducted study about such an under-researched topic like the lived experiences of children and adolescents with TB. The work’s focus on persistent sequelae, psychosocial implications, and post-TB aspirations among youth is relevant and well aligned with global health priorities. Study which has a child protection specialist during the IDI showed the authors inclination towards upholding principles of Child Rights and protection and must be definitely lauded as a great feat! These are few of my concerns and comments:

General comments:

1. Appreciate the detailed process of the study performed and the age-appropriate interventions and assessment tools used to understand the underlying concerns in children and adolescents. The use of IPA as methodology with participatory workshops was commendable.

2. Would recommend a simple clarification on how these participants were chosen so as to understand if any eligible participants declined and it was probably linked to poor experiences during TB care.

3. Would recommend a more detailed explanation of the various patterns seen in the pictographic representations of physical ailments seen.

4. A general description or sub group analysis to look at PTB Vs EPTB Vs Disseminated TB in the experiences would be useful to look at models to improve outcomes tailored by the type of TB encountered.

5. Would be advisable to clarify and tabulate participant and caregiver quotations separately as it becomes difficult to understand age based responses and perspectives.

6. Would recommend a small note on discussion of the various low cost high impact interventions you would suggest based on your findings to strengthen TB care in Gambia.

7. At most reflections, it is difficult to understand if the reflections are during the time of disease or treatment or now in Post TB life since some of the experiences are clouded by the concerns while on treatment ( drugs related, disease pathology related) which are rarely of any concerns post treatment. It may be imperative to speak about the lived experience as not just the post care period but also during the diagnosis, treatment and follow up period.

Looking forward to your edited manuscript and hopeful that the outcome will be definitely a good read!

**Do you want your identity to be public for this peer review?** For information about this choice, including consent withdrawal, please see our Privacy Policy

Reviewer #1: **Yes: ** Abdul Nazer Ali

Reviewer #2: **Yes: ** Nikith Austin DSouza

---

## [Editor Report · Decision Letter 1]

13 Nov 2025

"I live with pain, it cannot go away": lived experiences of childhood and adolescent pulmonary tuberculosis survivors - a qualitative study

PGPH-D-25-01458R1

Dear Dr Nkereuwem,

We are pleased to inform you that your manuscript '"I live with pain, it cannot go away": lived experiences of childhood and adolescent pulmonary tuberculosis survivors - a qualitative study' has been provisionally accepted for publication in PLOS Global Public Health.

Best regards,

Ferdinand C Mukumbang, PhD

Academic Editor